# Fabrication of Ag-CaCO_3_ Nanocomposites for SERS Detection of Forchlorfenuron

**DOI:** 10.3390/molecules28176194

**Published:** 2023-08-23

**Authors:** Fangyi Qin, Rongjun Liu, Qiong Wu, Shulong Wang, Fa Liu, Qingmin Wei, Jiayao Xu, Zhihui Luo

**Affiliations:** 1College of Pharmacy, Guangxi University of Chinese Medicine, Nanning 530001, China; 13481800591@163.com; 2Guangxi Key Laboratory of Agricultural Resources Chemistry and Biotechnology, Guangxi Colleges and Universities Key Laboratory for Efficient Use of Featured Resources in the Southeast of Guangxi, College of Chemistry and Food Science, Yulin Normal University, Yulin 537000, China; liu_rongjun@163.com (R.L.); 15235054973@163.com (Q.W.); shulonghs@163.com (S.W.); fa.l@163.com (F.L.); weiqingmin09@163.com (Q.W.); xujiayao_2011@163.com (J.X.)

**Keywords:** Ag-CaCO_3_ nanocomposites, synthesis, forchlorfenuron, SERS

## Abstract

In this study, Ag-CaCO_3_ nanocomposites were synthesized using silver nitrate as the precursor solution based on calcium carbonate nanoparticles (CaCO_3_ NPs). The synthesis involved the reaction of calcium lignosulphonate and sodium bicarbonate. The properties of Ag-CaCO_3_ nanocomposites were studied by various technologies, including an ultraviolet–visible spectrophotometer, a transmission electron microscope, and a Raman spectrometer. The results showed that Ag-CaCO_3_ nanocomposites exhibited a maximum UV absorption peak at 430 nm, the surface-enhanced Raman spectroscopy (SERS) activity of Ag-CaCO_3_ nanocomposites was evaluated using mercaptobenzoic acid (MBA) as the marker molecule, resulting in an enhancement factor of 6.5 × 10^4^. Additionally, Ag-CaCO_3_ nanocomposites were utilized for the detection of forchlorfenuron. The results demonstrated a linear relationship in the concentration range of 0.01 mg/mL to 2 mg/mL, described by the equation y = 290.02x + 1598.8. The correlation coefficient was calculated to be 0.9772, and the limit of detection (LOD) was determined to be 0.001 mg/mL. These findings highlight the relatively high SERS activity of Ag-CaCO_3_ nanocomposites, making them suitable for analyzing pesticide residues and detecting toxic and harmful molecules, thereby contributing to environmental protection.

## 1. Introduction

Calcium carbonate (CaCO_3_) is a crucial inorganic non-metallic substance widely present in nature, such as in marble, limestone, calcite, and Cretaceous deposits [1,2]. Owing to its abundant sources, economical nature, high degree of whiteness, as well as its non-toxic and tasteless properties, CaCO_3_ has found extensive utilization across diverse industries, encompassing daily chemical products, food, medicine, papermaking, rubber, and plastics [3,4,5]. As an innovative material for drug delivery, research has revealed that CaCO_3_ NPs exhibit advantages over alternative inorganic drug carriers, including safety, non-toxicity, excellent biocompatibility, notable porosity, expansive specific surface area, and swift degradation in mild environments. Consequently, CaCO_3_ NPs emerge as an ideal drug carrier [6,7,8,9,10,11]. As a result, the synthesis of multifunctional CaCO_3_ NPs has gained significant research attention within the realms of biomedicine, analytical chemistry, and environmental analysis.

Surface-enhanced Raman spectroscopy (SERS) is an analytical detection technology that offers numerous advantages over other techniques. These advantages can be summarized as follows. Firstly, SERS exhibits exceptional sensitivity, enabling the detection of single molecules. Secondly, it allows for the easy acquisition of spectral information from target molecules or groups, owing to its high selectivity, resolution, surface selection laws, and resonance enhancement. Thirdly, due to its non-destructive nature and field detection capabilities, SERS finds extensive applications in various fields, including drugs, food safety, diseases, explosives, imaging, minerals, and archaeology [12,13,14,15,16,17,18,19]. The achievement of ultra-high sensitivity in SERS relies on the utilization of precious metal nanomaterials. Consequently, there has been significant research focused on synthesizing nanomaterials with strong SERS activity, controllable particle size, excellent stability, and superior biocompatibility [20]. Advancements in nanoscience and technology have led to the synthesis of a variety of metal nanoparticles with diverse shapes, such as spherical, star, tetrahedral, icosahedral, nuclear shell, dendritic, and flower shapes, under controlled conditions. These metal nanoparticles, including gold–silver composites, have been utilized for direct detection or marking purposes [21,22,23,24]. From the existing literature, there are few reports about Ag-CaCO_3_ nanocomposites. Ahmed et al. dispersed Ag NCs on a nano-scale solid CaCO_3_ carrier by the seed-mediated thermal injection method, and Ag@CaCO_3_ nanocomposites showed excellent ultraviolet protection and anti-Escherichia coli activity [25]. Ueda et al. prepared calcium carbonate powder loaded with silver nanoparticles using an ultrasonic spray pyrolysis route, and Ag-CaCO_3_ obtained from 1% silver had minimal cytotoxicity and effective antibacterial activity [26].

In this study, we designed and synthesized CaCO_3_ NPs with uniform particle size, good stability, and water solubility. To achieve this, we employed calcium lignosulphonate and a modifier, optimizing their concentrations and reaction time. The carbonate ions for the formation of calcium carbonate were derived from sodium bicarbonate, and the resulting CaCO_3_ NPs acted as carriers, facilitating the electrostatic adsorption of Ag^+^ by utilizing sodium borohydride as a reducing agent, allowing us to successfully prepare Ag-CaCO_3_ nanocomposites. To assess the composite’s SERS activity, we employed mercaptobenzoic acid as a molecular marker, resulting in an enhancement factor of 6.5 × 10^4^. Finally, we demonstrated the application of the synthesized composite in the successful detection of chlorphenylurea. This method has low cost and simple operation. It can directly reduce silver nitrate in situ and control the particle size by adjusting the amount of silver nitrate, which also has a good enhancement effect and lays a foundation for the detection of other bacteria or substances.

## 2. Results and Discussion

### 2.1. Characterization of CaCO_3_ NPs

#### 2.1.1. Effects of Different Concentrations of Calcium Lignosulfonate on the Synthesized CaCO_3_ NPs

Through an examination of the impacts stemming from diverse concentrations of calcium lignosulfonate on the synthesized CaCO_3_ NPs, as illustrated in Figure 1A, it becomes apparent that the hue of the resulting product progressively intensified with an escalation in calcium lignosulfonate concentration. Laser irradiation showcased a luminous pathway within the synthesized product, indicating successful dispersion. Nevertheless, starting from the 4th sample, the laser lights gradually grew dimmer, implying inadequate transmittance and the existence of particle precipitation. Based on these observations, it was concluded that 0.004 mol/L of calcium lignosulfonate represents the optimal concentration for the synthesis of CaCO_3_ NPs.

#### 2.1.2. TEM Images of CaCO_3_ NPs

The morphology of the synthesized CaCO_3_ NPs was investigated through TEM characterization of the synthesized materials, and the findings are presented in Figure 2. As the concentration of calcium lignosulfonate increased, the synthesized CaCO_3_ NPs displayed varying sizes, transitioning from aggregation to dispersion and subsequently returning to aggregation. Upon the addition of 0.004 mol/L of calcium lignosulfonate, the carbonates fused together uniformly, forming spherical structures due to the modifying effect of calcium lignosulfonate. The synthesized CaCO_3_ NPs exhibited particle sizes ranging from 150–200 nm, and the spherical calcium carbonate exhibited a consistent morphology. This observation implies that a low dosage of calcium lignosulfonate may lead to an excessive presence of sodium hydrogen carbonate, resulting in the insignificant modification of the CaCO_3_ NPs and uneven particle sizes. Conversely, an excessively high dose of calcium lignosulfonate may induce the adsorption and clustering of calcium carbonate particles by lignosulfonate, impeding the formation of spherical calcium carbonate particles.

#### 2.1.3. Infrared Spectra of CaCO_3_ NPs

The role of calcium lignosulphonate in the synthesis of CaCO_3_ NPs was investigated using infrared spectra for characterization. The results are presented in Figure 3. Curves a and b exhibit stretching vibration peaks at 3431.9 cm^−1^, 1112.2 cm^−1^, and 622.7 cm^−1^. The peak at 3431.9 cm^−1^ corresponds to the stretching vibration absorption peak of O–H bonds. The peak at 1112.2 cm^−1^ is attributed to the stretching vibration of C–O bonds in the aromatic ring. The peak at 876.1 cm^−1^ comes from the out-of-plane deformation vibration of CO_3_^2−^. Additionally, the prominent peak at 622.7 cm^−1^ mainly arises from the stretching vibration of the sulfonic acid group (S=O). This demonstrates that calcium lignosulfonate can be utilized as a calcium source and a modifier for calcium carbonate nanoparticles during their synthesis.

#### 2.1.4. X-ray Diffraction (XRD) Spectra of CaCO_3_ NPs

X-ray diffraction (XRD) analysis was performed to determine the crystalline structure of the synthesized CaCO_3_ NPs. The results are shown in Figure 4. Seven distinct diffraction peaks were observed at diffraction angles of 23.1°, 29.7°, 36.1°, 39.3°, 43.1°, 47.4°, and 48.5°, respectively. The corresponding crystal planes are as follows (012), (104), (110), (113), (202), (018), and (116). A comparison of the diffraction pattern of CaCO_3_ NPs with that of the standard CaCO_3_ sample reveals similar diffraction peaks, although with slight variations in the positions of the characteristic peaks. Notably, no impurity peaks are detected, indicating the relatively high purity of the sample. The XRD analysis confirms that the synthesized CaCO_3_ NPs have a crystal form identified as calcite.

#### 2.1.5. Solubility Research of CaCO_3_ NPs

The laser beam technique was employed to investigate the water solubility and stability of the synthesized CaCO_3_ NPs. Figure 5 shows that the observed light paths of the samples remained consistently clear and uniform for 5 days with minimal changes. This indicates that the synthesized CaCO_3_ NPs exhibit excellent solubility in water and maintain a good dispersion. Therefore, the synthesized CaCO_3_ NPs hold great potential for wide-ranging applications in fields such as biomedicine, analytical chemistry, and environmental analysis.

### 2.2. Characterization of the Ag-CaCO_3_ Nanocomposites

#### 2.2.1. Effects of Different Concentrations of AgNO_3_ on the Synthesized Ag-CaCO_3_ Nanocomposites

In order to investigate the effects of different concentrations of AgNO_3_ on the synthesized Ag-CaCO_3_ nanocomposites, UV-vis spectrometry was employed. Figure 6 displays the UV-vis spectra of the synthesized Ag-CaCO_3_ nanocomposites under different concentrations of AgNO_3_. The blank group (line e) exhibited a noticeable absorption peak at 388 nm, which corresponds to the surface plasmon resonance of silver nanoparticles. Typically, the UV peak for silver nanoparticles occurs between 380 and 420 nm. With an increase in silver nitrate concentration, the maximum absorption peak of the synthesized Ag-CaCO_3_ nanocomposites shifted to around 435 nm while its intensity steadily increased. Furthermore, as the silver nitrate content increased, the maximum absorption peak broadened, suggesting the potential occurrence of particle agglomeration within the synthesized materials. Therefore, the optimal concentration of silver nitrate for synthesizing the Ag-CaCO_3_ nanocomposites was found to be 0.003 mol/L.

Concurrently, the synthesized Ag-CaCO_3_ nanocomposites were characterized using TEM. Figure 7A depicts the smooth surface of CaCO_3_ NPs with a particle size of roughly 150 nm. Furthermore, as the concentration of AgNO_3_ increased (shown in Figure 7B–D), an increasing number of minute black nanoparticles were observed to be modified on the CaCO_3_ NPs surface. The average particle size of silver nanoparticles was 20.53 nm as measured by Image J. Subsequently, these modified Ag-CaCO_3_ nanocomposites were utilized for SERS detection.

#### 2.2.2. SERS Activity Study of Ag-CaCO_3_ Nanocomposites

Mercaptobenzoic acid (MBA) was employed as a marker molecule to assess the SERS activity of the synthesized Ag-CaCO_3_ nanocomposites and explore its potential applications. Thorough mixing of 1.0 mL of the synthesized Ag-CaCO_3_ nanocomposites with 3 μL of MBA (10^−3^ M) was performed. Subsequently, a 30 min incubation was followed by the drop-casting of 3 μL of the mixture onto a clean piece of tin foil, which was then air-dried for subsequent SERS analysis. The SERS measurements were conducted under the following experimental conditions: a 785 nm laser was utilized as the excitation source, the sample was irradiated at a laser power of 20%, the exposure time was 3 s, and three accumulations were performed. Figure 8 illustrates the significant SERS bands at 1076 and 1587 cm^−1^, corresponding to the ν12 and ν8a aromatic ring vibrations, respectively [27,28]. The SERS activity of Ag-CaCO_3_ nanocomposites was calculated using the equation EF = (I_SERS_/I_Raman_)/(C_SERS_/C_Raman_) to estimate the enhancement factor (EF) [29], Where I_SERS_ and I_Raman_ are the intensity of the ν_8a_ (1587 cm^−1^) in the SERS and normal Raman spectra of the solution. Ag-CaCO_3_ nanocomposites exhibited a calculated EF value of 6.5 × 10^4^, indicating a significant enhancement in the Raman signal. This exceptional SERS activity suggests that the composite holds great potential for the detection of biomacromolecules and environmental molecules.

### 2.3. SERS Detection of Forchlorfenuron

Forchlorfenuron acts as a plant growth regulator that stimulates the growth and development of plants, thereby improving crop yield and quality. However, the presence of forchlorfenuron residues can pose chronic toxicity to the human body, potentially causing long-term damage. Additionally, it can contribute to environmental pollution and disrupt the ecological balance. To assess the applicability of the synthesized Ag-CaCO_3_ nanocomposites, we conducted a direct test on forchlorfenuron using Ag-CaCO_3_ nanocomposites as a SERS substrate.

The spectra obtained from analyzing various concentrations of forchlorfenuron using the Ag-CaCO_3_ nanocomposites as the SERS substrate are presented in Figure 9. It is evident that as the concentration of forchlorfenuron decreases, the SERS peak gradually diminishes. A plot was generated to establish a correlation between the intensity of the characteristic absorption peak at 988 cm^−1^ and the concentration of forchlorfenuron. As shown in Figure 10, this resulted in a linear equation: y = 290.02x + 1598.8 (R^2^ = 0.9624), where x represents the forchlorfenuron concentration and y represents the SERS strength of the target substance. The limit of detection (LOD) was determined to be 0.001 mg/mL, the lowest concentration reliably detected by the SERS method.

## 3. Experiment

### 3.1. Experiment Reagents

Calcium lignosulphonate (96%) and sodium bicarbonate (AR) were purchased from Shanghai Yuanye Biotech Co., Ltd. (Shanghai, China). Sodium borohydride (AR, NaBH_4_), silver nitrate (AR, AgNO_3_), hydroxylamine hydrochloride (AR, NH_3_OHCl), and ascorbic acid (AR, C₆H₈O₆) were obtained from Sinopharm Group Chemical Reagent Co., Ltd. (Shanghai, China).

### 3.2. Experimental Apparatus

UV-vis spectrophotometer (Shimadzu Instrument Co., Ltd., Kyoto, Japan), G220 Tecnai transmission electron microscope (TEM) (FEI Company, Hillsboro, OR, USA), and Fourier transform infrared spectrometer (PE Company, Waltham, MA, USA) were applied. A high-resolution micro-Raman spectrometer (Xiamen Pushi Nanotechnology Co., Ltd., Xiamen, China) and D8 advance X-ray diffraction (Germany Bruker Co., Ltd., Bremen, Germany) were used.

### 3.3. Synthesis of CaCO_3_ NPs

To prepare the CaCO_3_ NPs, 5 mL of a 0.02 M sodium hydrogen carbonate solution was collected and transferred into a 30 mL glass bottle with a cover. Then, a specified volume of 0.02 M calcium lignosulfonate solution was slowly added under continuous stirring. The reaction was allowed to proceed for 30 min, followed by centrifugation to separate the particles. The CaCO_3_ NPs were then redispersed using 5 ml of deionized water.

### 3.4. Synthesis of Ag-CaCO_3_ Nanocomposites

To prepare the Ag-CaCO_3_ nanocomposites, 2 mL of CaCO_3_ NPs suspension solution was transferred into a 30 mL bottle. Deionized water was added to dilute the suspension to a total volume of 10 mL. The mixture was subjected to an ultrasound treatment for 10 min while being continuously stirred. During the process, a suitable volume of a 0.01 M silver nitrate solution was added dropwise. After 30 min of reaction, centrifugation was performed. The resulting pellet was then dispersed using deionized water, and 200 µL of 1% sodium citrate was added while stirring continuously. Additional silver nitrate was added, and silver nitrate was immediately reduced using sodium borohydride. The reaction was terminated after 30 min. To obtain the final composite, the solution was centrifuged for 10 min at a rate of 6000 rpm. Through these steps, the Ag-CaCO_3_ nanocomposites were successfully prepared.

### 3.5. SERS Detection of Forchlorfenuron

To investigate the SERS performance of the Ag-CaCO_3_ nanocomposites, different concentrations of forchlorfenuron (2 mg/mL, 1 mg/mL, 0.1 mg/mL, 0.01 mg/mL, 0.001 mg/mL, and 0.0001 mg/mL) were added to 1 mL of the Ag-CaCO_3_ nanocomposites. The mixture was allowed to react at room temperature for 1 h. Next, a 3 μL aliquot of the resulting mixture was carefully dropped onto a piece of clean tin foil and left to dry in ambient air. The SERS measurements were conducted under the following conditions: a 785 nm laser was used as the light source, with a laser power of 20% for sample irradiation. The exposure time was set to 3 s, and the accumulation times were determined to be 3.

## 4. Conclusions

In this study, we successfully synthesized CaCO_3_ NPs exhibiting uniform particle size, commendable stability, and water solubility by incorporating modifiers and meticulous optimization of the concentration and reaction time of the calcium source, namely calcium lignosulfonate. Furthermore, Ag-CaCO_3_ nanocomposites were prepared by introducing silver nitrate and employing sodium borohydride as the reducing agent. The resulting Ag-CaCO_3_ nanocomposites exhibited exceptional SERS performance and proved effective in the detection of forchlorfenuron, with a remarkably low limit of detection of 0.001 mg/mL. The examination of the Ag-CaCO_3_ nanocomposites further established a robust linear relationship with forchlorfenuron. These experimental findings underscore the promising potential of the synthesized Ag-CaCO_3_ nanocomposites in the realm of SERS applications.

## Figures and Tables

**Figure 1 molecules-28-06194-f001:**
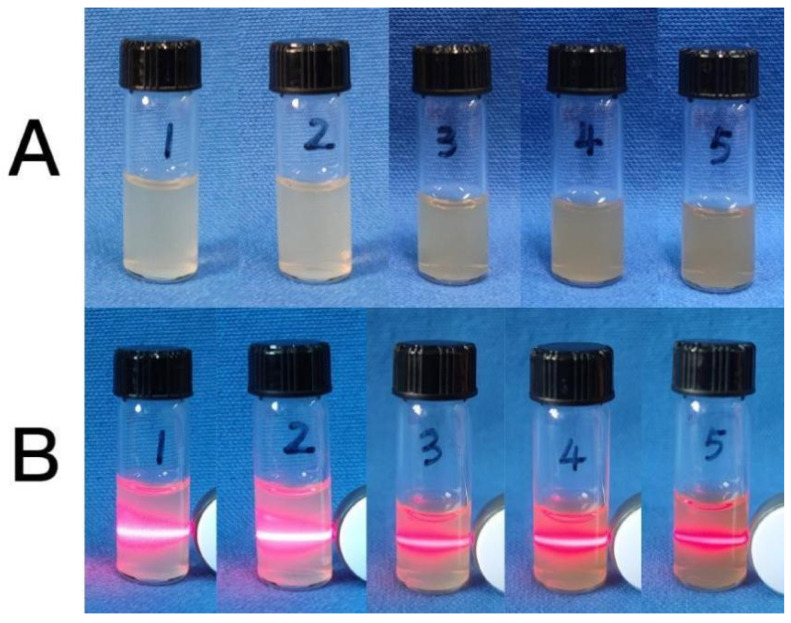
(**A**) Photo of the synthesized CaCO_3_ NPs at different concentrations of calcium lignosulfonate solution; (**B**) Photograph of illuminated with a laser pen (1: 0.0004 mol/L, 2: 0.001 mol/L, 3: 0.004 mol/L, 4: 0.008 mol/L, 5: 0.012 mol/L, calcium lignosulfonate).

**Figure 2 molecules-28-06194-f002:**
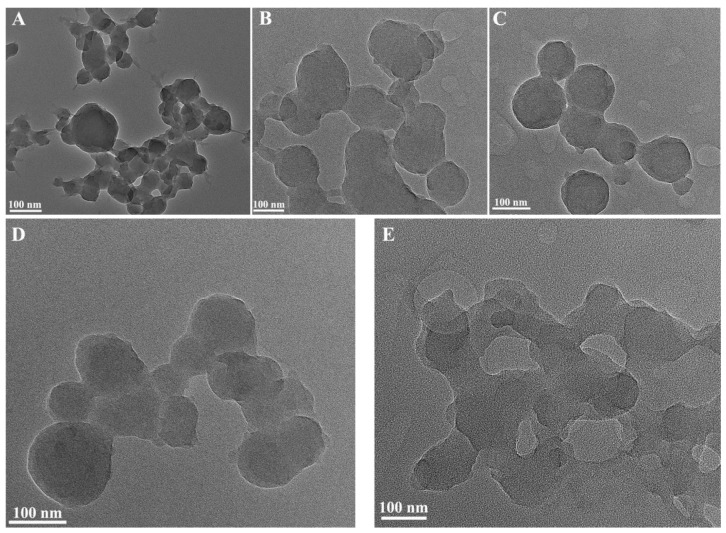
TEM of the synthesized CaCO_3_ NPs at different concentrations of calcium lignosulfonate solution (**A**–**E**) 0.0004 mol/L, 0.001 mol/L, 0.004 mol/L, 0.008 mol/L, 0.012 mol/L calcium lignosulfonate).

**Figure 3 molecules-28-06194-f003:**
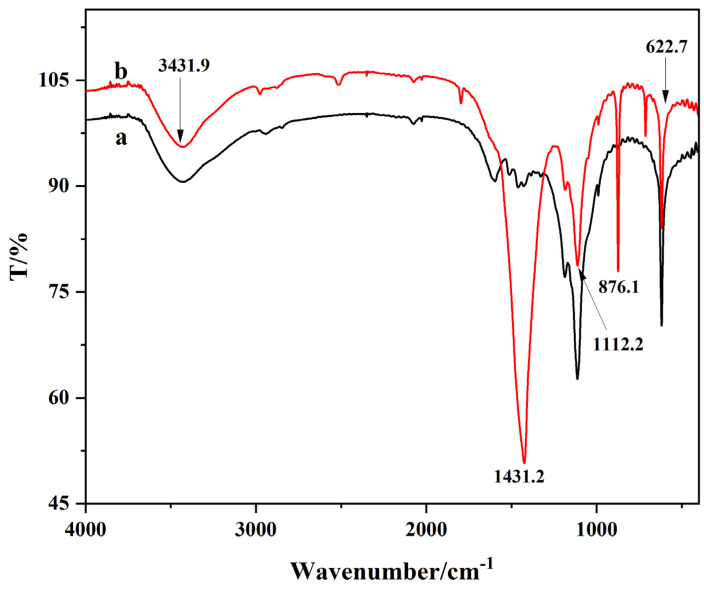
Infrared spectrum of the synthesized calcium lignosulphonate (a) and CaCO_3_ NPs (b).

**Figure 4 molecules-28-06194-f004:**
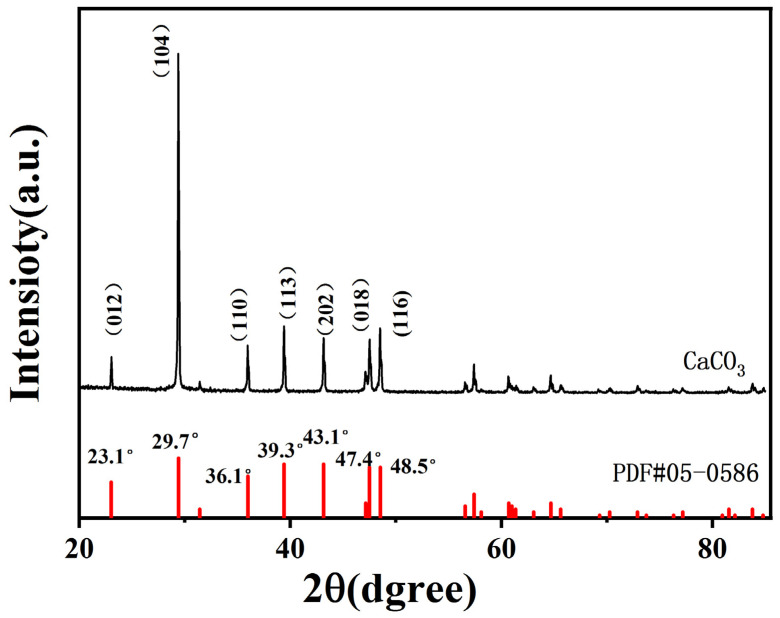
X-ray diffraction pattern of the synthesized CaCO_3_ NPs.

**Figure 5 molecules-28-06194-f005:**
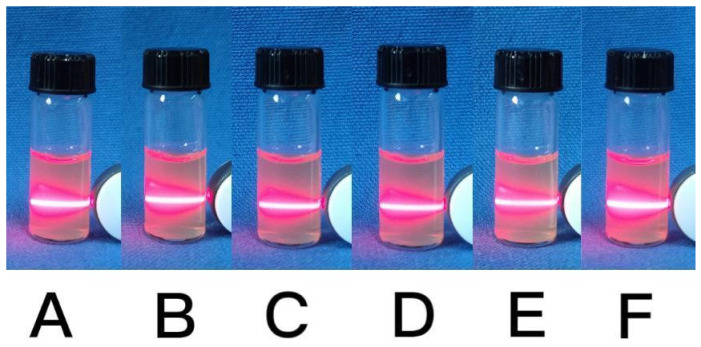
Photos of stability of the synthesized CaCO_3_ NPs at different times (**A**) 0 h, (**B**) 4 h, (**C**) 12 h, (**D**) 24 h, (**E**) 72 h, (**F**) 120 h.

**Figure 6 molecules-28-06194-f006:**
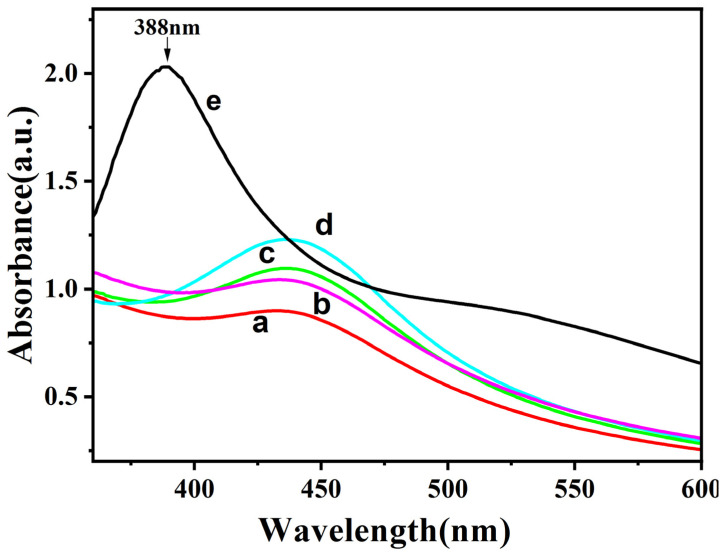
UV-visible absorption spectra of the synthesized Ag-CaCO_3_ nanocomposites under different concentrations of AgNO_3_. ((a–d) 0.001 mol/L, 0.002 mol/L, 0.003 mol/L, 0.004 mol/L AgNO_3_; (e) 0.002 mol/L AgNO_3_, calcium carbonate free).

**Figure 7 molecules-28-06194-f007:**
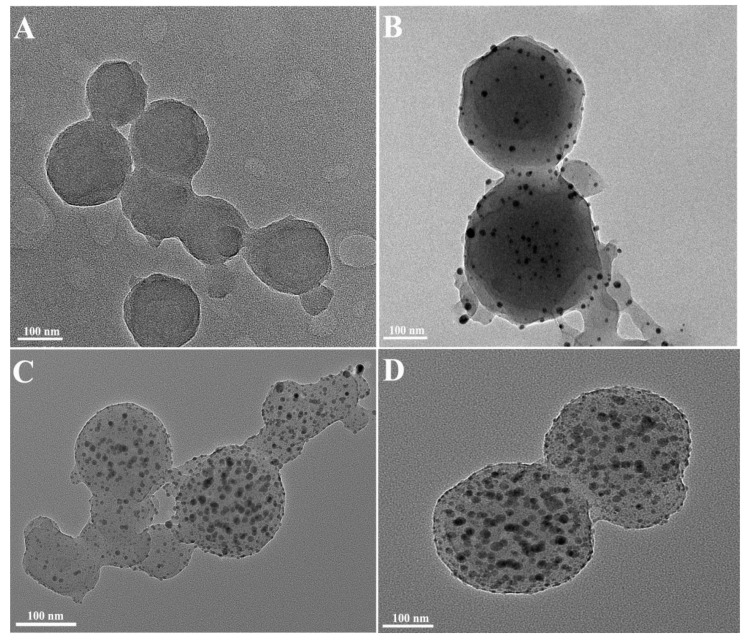
TEM images of the obtained Ag-CaCO_3_ nanocomposites by adding different amounts of silver nitrate. (**A**–**D**) 0 mol/L, 0.002 mol/L, 0.003 mol/L, and 0.004 mol/L AgNO_3_.

**Figure 8 molecules-28-06194-f008:**
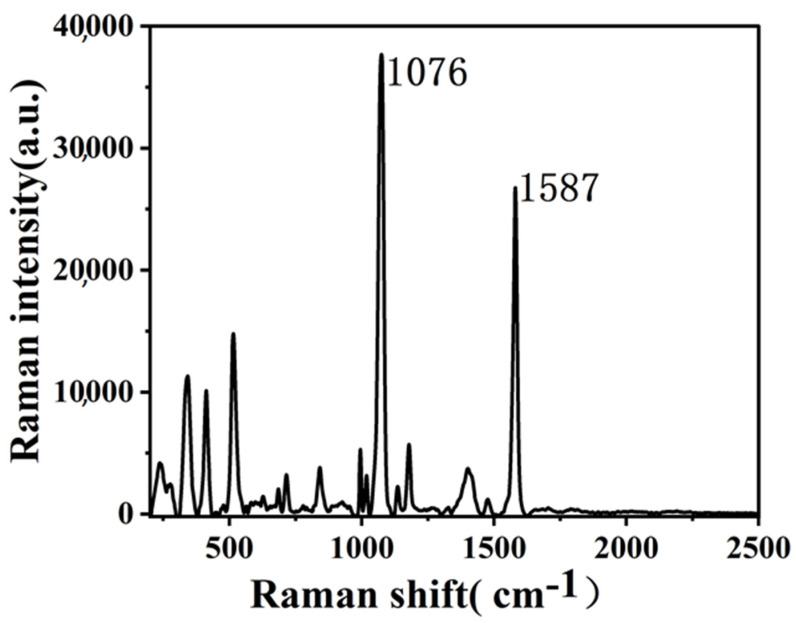
Surface-enhanced Raman spectra of MBA on the obtained Ag-CaCO_3_ nanocomposites.

**Figure 9 molecules-28-06194-f009:**
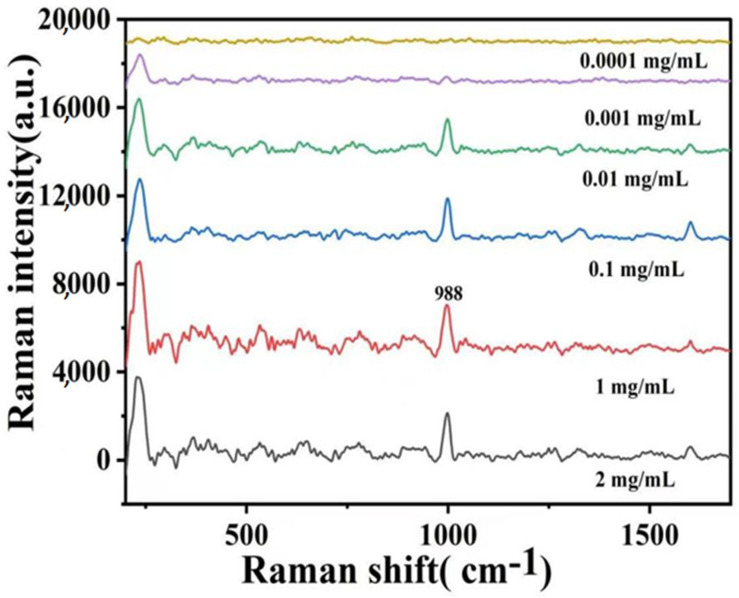
Surface-enhanced Raman spectra of forchlorfenuron with different concentrations of the obtained Ag-CaCO_3_ nanocomposites.

**Figure 10 molecules-28-06194-f010:**
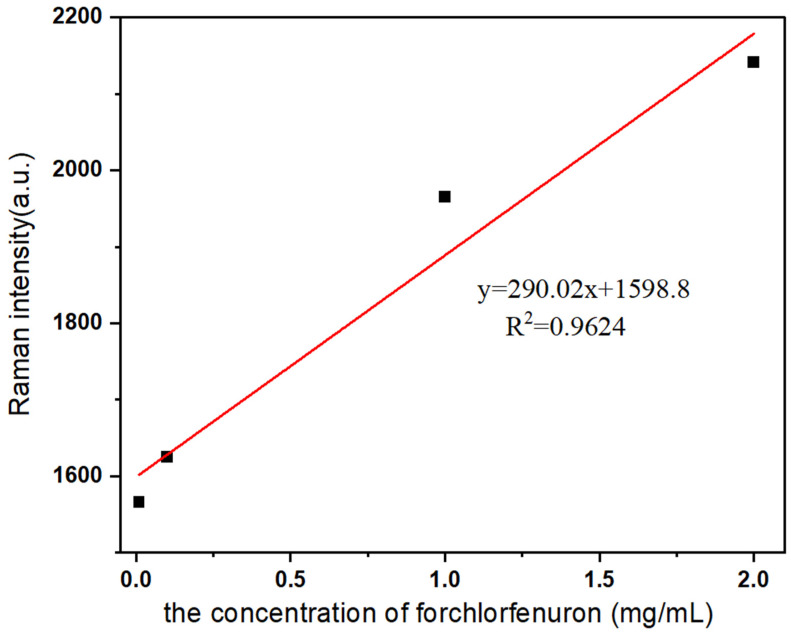
A plot between the SERS intensity at 988 cm^−1^ and the concentration of forchlorfenuron.

## Data Availability

Not applicable.

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
