# Peer review of "Fabrication of Ag-CaCO3 Nanocomposites for SERS Detection of Forchlorfenuron"

_molecules, 2023, doi:10.3390/molecules28176194_

Round 1
Reviewer 1 Report
The present manuscript entitled “Synthesis of Ag-CaCO3 nanoparticles and its application for the detection of forchlorfenuron with SERS” by Qin et al., describes the preparation of Ag-CaCO3 nanoparticles (SCN) by using silver nitrate as the precursor solution based on calcium carbonate nanoparticles (CCN). Characterization analysis of as-synthesized SCN was performed using various techniques such as XRD, UV-Vis, Raman, and TEM analysis. Furthermore, the findings of the current study highlight the relatively high surface-enhanced Raman spectroscopy (SERS) activity of SCN. The authors report an interesting work. The objective and justification of the work are clear. Therefore, I recommend it for publication. However, some issues are detailed below which need to be addressed before its final acceptance in Molecules.
I advise the authors to take the following points into account while revising their manuscript.
Comment 1: There are some typographical and grammatical errors in the manuscript text, so the authors need to correct them in the revised manuscript.
Comment 2: The whole manuscript must be cross-checked thoroughly for English editing, grammatical, spelling mistakes, and syntax errors. So, I suggest the author's English language should be polished.
Comment 3: Keywords are missing in the manuscript, so it need to be added in the revised manuscript.
Comment 4: Include some more recent references in the introduction to strengthen the section.
Comment 5: Include all chemicals purity in section 2.1 Experiment and reagents.
Comment 6: In section 2.2 Experimental apparatus, XRD and RAMAN instrumental details are missing. So include them in the revised manuscript.
Comment 7: The Figure 1 caption needs to be revised as mentioned in the Figure A and B.
Comment 8: Revise the X-axis scale of Figure 3 from 400-4000 cm-1.
Comment 9: Calculate the crystallite size using XRD analysis and compare the size with TEM analysis. Include the Miller indices in the XRD analysis.
Comment 10: Figure 6 axis titles are looking stretched and blurry, so rewrite the axis title with high-resolution.
Comment 11: In line 185, the authors mentioned that “AgNO3, it can be seen that more and more small black nanoparticles were modified on the surface of CCN” So, I suggest the authors mention the size of the black nanoparticles using Image J software.
Comment 12: Regarding the conclusions section, include clear quantitative findings and more emphasis on the findings and its implication may be mentioned in the conclusion section, also the conclusion section needs to be elaborated.
Comment 13: The homogeneity of the reference section needs to be maintained. In some references, the journal names are written full form, and some are in abbreviated form (Reference 8). So please check and revise accordingly to the journal's instructions.

Moderate editing of English language required.
Author Response
Dear reviewers,
On behalf of my co-authors, we thank you very much for giving us an opportunity to revise our manuscript, we appreciate editor and reviewers very much for their positive and constructive comments and suggestions on our manuscript entitled “Fabrication of Ag-CaCO3 nanohybrid for SERS detection of forchlorfenuron” (Molecules-2493104). We have tried our best to revise our manuscript according to the comments, and the revised portion has been marked in red color in the revised manuscript.
The main corrections in the paper and the responds to the reviewer’s comments are shown in the additive list below.
Thank you and looking forward to hearing from you.
Sincerely yours,
Zhihui Luo
Jul 13, 2023
To reviewer #1:
Comment 1: There are some typographical and grammatical errors in the manuscript text, so the authors need to correct them in the revised manuscript.
Answer: Thank you for your good suggestions. We have revised the manuscript carefully by English teacher and marked in red color in the revised manuscript.
Comment 2: The whole manuscript must be cross-checked thoroughly for English editing, grammatical, spelling mistakes, and syntax errors. So, I suggest the author's English language should be polished.
Answer: Thank you for your good suggestions. We have polished the manuscript carefully by experienced colleagues and English teacher and marked in red color in the revised manuscript.
Comment 3: Keywords are missing in the manuscript, so it need to be added in the revised manuscript.
Answer: Thanks a lot for your valuable comments. We have added keywords in the revised manuscript. Such as: Ag-CaCO3 nanohybrid; synthesis; forchlorfenuron; SERS.
Comment 4: Include some more recent references in the introduction to strengthen the section.
Answer: Thank you very much for your valuable comments. We have quoted more recent references in the revised manuscript.
Comment 5: Include all chemicals purity in section 2.1 Experiment and reagents.
Answer: Thanks a lot. We have added the corresponding purity after the chemicals.
Comment 6: In section 2.2 Experimental apparatus, XRD and RAMAN instrumental details are missing. So include them in the revised manuscript.
Answer: According to the kind advice of the reviewer, we have supplemented the relevant information of X-ray diffraction and high-resolution micro-Raman spectrometer in Section 2.2.
Comment 7: The Figure 1 caption needs to be revised as mentioned in the Figure A and B.
Answer: Thank you very much for your valuable comment. We have revised the caption of Figure 1. Such as: Figure 1. A: Photo of the synthesized CaCO3 NPs at different concentrations of calcium lignosul-fonate solution ; B: Photograph of illuminated with a laser pen (1: 0.0004mol/L,2: 0.001mol/L,3: 0.004mol/L,4: 0.008mol/L,5: 0.012mol/L, calcium lignosulfonate).
Comment 8: Revise the X-axis scale of Figure 3 from 400-4000 cm-1.
Answer: Thanks a lot for your valuable Comment. We have adjusted the scale of the X axis in Figure 3 from 400-4000cm-1.
Figure 3. Infrared spectrum of the synthesized calcium lignosulphonate (a) and CaCO3 NPs (b).
Comment 9: Calculate the crystallite size using XRD analysis and compare the size with TEM analysis. Include the Miller indices in the XRD analysis.
Answer:
Thank you for your good suggestions. We have added the Miller indices analysis in the revised manuscript and in marked in red color.
Figure 4. X-ray diffraction pattern of the synthesized calcium carbonate nanoparticles.
Comment 10: Figure 6 axis titles are looking stretched and blurry, so rewrite the axis title with high-resolution.
Answer: According to the kind advice of the reviewer, we have modified the axis title in Figure 6.
Figure 6. UV-visible absorption spectra of the synthesized Ag-CaCO3 nanohybrid under different concentration of AgNO3. (a-d:0.001 mol/L, 0.002 mol/L, 0.003 mol/L, 0.004 mol/L AgNO3; ; e: 0.002 mol/L AgNO3, Calcium carbonate free).
Comment 11: In line 185, the authors mentioned that “AgNO3, it can be seen that more and more small black nanoparticles were modified on the surface of CCN” So, I suggest the authors mention the size of the black nanoparticles using Image J software.
Answer: Thanks a lot for your valuable advice. We used Image J to measure silver nanoparticles and supplemented the corresponding data in the article.
Comment 12: Regarding the conclusions section, include clear quantitative findings and more emphasis on the findings and its implication may be mentioned in the conclusion section, also the conclusion section needs to be elaborated.
Answer: Thanks a lot for your valuable comment, which we explained in more detail in the conclusion. Such as: In this study, we successfully synthesized CaCO3 NPs exhibiting uniform particle size, commendable stability, and water solubility through the incorporation of modifiers and meticulous optimization of the concentration and reaction time of the calcium source, namely calcium lignosulfonate. Furthermore, Ag-CaCO3 nanohybrid was prepared by in-troducing silver nitrate and employing sodium borohydride as the reducing agent. The resulting Ag-CaCO3 nanohybrid exhibited exceptional SERS performance and proved ef-fective in the detection of forchlorfenuron, with a remarkably low limit of detection of 0.001 mg/mL. The examination of the Ag-CaCO3 nanohybrid further established a robust linear relationship with forchlorfenuron. These experimental findings underscore the promising potential of the synthesized Ag-CaCO3 nanohybrid in the realm of SERS applications.
Comment 13: The homogeneity of the reference section needs to be maintained. In some references, the journal names are written full form, and some are in abbreviated form (Reference 8). So please check and revise accordingly to the journal's instructions.
Answer: According to the kind suggestions of the reviewer, we adjusted the references.
Reviewer 2 Report
Manuscript ID: molecules-2493104
Title: Synthesis of Ag-CaCO3 nanoparticles and its application for the detection of forchlorfenuron with SERS
The authors described the synthesis and characterization of Ag-CaCO3 composites, with application in the detection of forchlorfenuron by SERS technique.
In my opinion, the manuscript is not well-written, and the experiments and their interpretation must be improved. Some statements have no justification and an inappropriate scientific language was sometimes used. In consequence, I can not recommend the publication of manuscript.
1. The title – “Synthesis of Ag-CaCO3 nanoparticles and its application” – Is it about the application of “synthesis” or the application of “nanoparticles”?
2. Keywords are missing.
3. The acronym “SCN” must be replaced, because “SCN” is the chemical formula for thiocyanate, so the notation is reserved and can be confusing.
4. The state-of-the-arts for Ag-CaCO3 composites must be described in “Introduction”.
5. The novelty / originality of the proposed synthesis method must be stated.
6. The synthesis of composite must be described so that it can be reproduced by readers, with precise information.
7. The characterization of obtained materials is not satisfactory and the interpretation of results must be improved.
Some specific observations are the following:
8. “The particle size of the synthesized CCN ranged from 150-200 nm” – So they are not nanoparticles. If the authors do not have experimental evidence for belonging to the nano domain, the term "nanoparticles" and the abbreviation "CCN" must be removed.
9. XRD, FTIR, SEM, and EDX are necessary for the characterization of “SCN”.
10. Fig. 2 – The TEM images are not adequate. SEM images are necessary.
11. 3.1.3. Infrared spectra of CCN – The bands assigned to carbonate ion must be evidenced. It does not appear from the text that the FTIR spectra proved the synthesis of calcium carbonate.
12. The XRD pattern (Fig. 4) must be indexed.
13. The chemical formula of calcium carbonate can be use instead of “CC”. It is a chemistry journal though.
14. The affirmation “It is considered an ideal drug carrier” must be sustained by more arguments than a cited article.
15. “In this study, we designed and synthesized CCN with uniform particle size, good stability and high water solubility” – High water solubility?
16. “2.1. Experiment reagents” – The chemical formulas of reagents are missing.
17. “2.2. Experimental apparatus” – The names of analytical balance and magnetic stirrer are not necessary.
18. “To prepare the CCN solution” – Is it a true solution?
19. “The CCN were then redispersed in an appropriate amount of secondary water.” – Which is the “appropriate amount”? What is “secondary water”?
20. “Ag-CaCO3 nanoparticles” must be “Ag-CaCO3 nanocomposites” (if Ag particles are in the nano range).
21. “an appropriate amount of CCN suspension”, “a suitable volume”, “additional silver nitrate”, etc. – the quantities/ volumes must be provided.
22. “the mixture was reduced using sodium borohydride” – The whole mixture was reduced?
23. “1 mL of the SCN” – SCN is not a liquid.
24. “a 2 mL solution of calcium lignosulphonate” – the concentration must be given.
25. “a 2 mL solution of calcium lignosulphonate was determined to be the optimal concentration for the synthesized CCN” – Why “concentration” for a volume?
26. Figure 1. – They are different volumes, so different quantities, not “different concentrations”.
27. “2 mL of calcium lignosulphonate solution” – the concentration is missing.
28. “the carbonates were uniformly combined” – What “carbonates”?
29. “the optimal concentration of AgNO3 for the synthesis of the SCN was determined to be 3 mL” – “3 mL” is not a “concentration”.
30. Fig. 10 – They are only 4 points, more are needed.
31. Some examples of inappropriate scientific language: “Characterization analysis”, “Sodium hydrogen carbonate was used to introduce carbonates into the system.”, etc.
Author Response
Dear reviewer,
On behalf of my co-authors, we thank you very much for giving us an opportunity to revise our manuscript, we appreciate editor and reviewers very much for their positive and constructive comments and suggestions on our manuscript entitled “Fabrication of Ag-CaCO3 nanohybrid for SERS detection of forchlorfenuron” (Molecules-2493104). We have tried our best to revise our manuscript according to the comments, and the revised portion has been marked in red color in the revised manuscript.
The main corrections in the paper and the responds to the reviewer’s comments are shown in the additive list below.
Thank you and looking forward to hearing from you.
Sincerely yours,
Zhihui Luo
Jul 13, 2023
Reviewer #2:
Comment 1. The title – “Synthesis of Ag-CaCO3 nanoparticles and its application” – Is it about the application of “synthesis” or the application of “nanoparticles”?
Answer: We were sorry that our expression is unscientific. We have revised the title in the revised manuscript. Such as: Fabrication of Ag-CaCO3 nanohybrid for SERS detection of forchlorfenuron.
Comment 2. Keywords are missing.
Answer: Thank you very much for your valuable comment. We have added keywords to the article. Keywords: Ag-CaCO3 nanohybrid; synthesis; forchlorfenuron; SERS
Comment 3. The acronym “SCN” must be replaced, because “SCN” is the chemical formula for thiocyanate, so the notation is reserved and can be confusing
Answer: According to the kind suggestionof the reviewer, We have completely replaced "SCN" with Ag-CaCO3 nanohybrid.
Comment 4. The state-of-the-arts for Ag-CaCO3 composites must be described in “Introduction”.
Answer: Thanks a lot for your valuable comment. We have added some latest technologies of Ag-CaCO3 nanocomposites in the introduction, please refer to the reference (Barhoum et al., 2016, ACS applied materials & interfaces. Ueda et al., 2021, Materials Science and Engineering: c)
Comment 5. The novelty / originality of the proposed synthesis method must be stated.
Answer: Thank you very much for your valuable advice. We have added the novelty of the synthesis method of Ag-CaCO3 nanohybrid at the end of the article introduction.
Comment 6. The synthesis of composite must be described so that it can be reproduced by readers, with precise information.
Answer: According to the kind advice of the reviewer, we described the synthesis process of Ag-CaCO3 nanohybrid more accurately.
Comment 7. The characterization of obtained materials is not satisfactory and the interpretation of results must be improved. Some specific observations are the following:
Answer: Thank you for your good suggestions. We have carefully polished the manuscript and discussed specific observations, and marked in red color in the revised manuscript.
Comment 8. “The particle size of the synthesized CCN ranged from 150-200 nm” – So they are not nanoparticles. If the authors do not have experimental evidence for belonging to the nano domain, the term "nanoparticles" and the abbreviation "CCN" must be removed.
Answer: Thank you for your good suggestions. The abbreviation "CCN" has been replaced by CaCO3 NPs and the synthesized CaCO3 NPs ranged from 150-200 nm were synthesized under the different concentrations of calcium lignosulfonate. Such as the 200nm of CCN were obtained while the concentrations of calcium lignosulfonate is 0.004mol/L.
Comment 9. XRD, FTIR, SEM, and EDX are necessary for the characterization of “SCN”.
Answer: Thank you for your good suggestions, we have characterized the obtained nanoparticles with UV, XRD, FTIR ,TEM, Raman Spectroscopy. SEM, and EDX were not used to characterize the obtained nanoparticles because the machine didn’t work for 6 months, we have been waiting for it to be repaired.
Comment 10. Fig. 2 – The TEM images are not adequate. SEM images are necessary.
Answer: Thank you for your good suggestions, we also want to provide a good SEM, unfortunately, the machine didn’t work for 6 months in our laboratory, we have been waiting for it to be repaired.
Comment 11. 3.1.3. Infrared spectra of CCN – The bands assigned to carbonate ion must be evidenced. It does not appear from the text that the FTIR spectra proved the synthesis of calcium carbonate.
Answer: Thanks a lot for your valuable advice. The infrared spectrum of calcium carbonate was previously annotated incorrectly, and now the spectrum has been revised. the peak at 876.1cm-1 comes from the out-of-plane deformation vibration of CO32-.
Comment 12. The XRD pattern (Fig. 4) must be indexed.
Answer: According to the kind suggestion of the reviewer, we have added the standard card PDF#05-0586 of calcium carbonate to the XRD pattern.
Comment 13. The chemical formula of calcium carbonate can be use instead of “CC”. It is a chemistry journal though.
Answer: Thank you very much for your valuable advice. CC has been replaced CaCO3 in the revised manuscript.
Comment 14. The affirmation “It is considered an ideal drug carrier” must be sustained by more arguments than a cited article.
Answer: Thanks a lot for your valuable comment. We have added more references to the article, please refer to the reference (Dong et al, 2020, Chem; Dong et al, 2020, Nano Research; Xu et al, 2019, Advanced Functional Materials; Xu et al, 2022, ACS Applied Bio Materials, Wang et al, 2022, Advanced Materials).
Comment 15. “In this study, we designed and synthesized CCN with uniform particle size, good stability and high water solubility” – High water solubility?
Answer: Thank you very much for your valuable comment. We have revised the improper description.
Comment 16. “2.1. Experiment reagents” – The chemical formulas of reagents are missing.
Answer: According to the well-intentioned suggestion of the reviewer, we have added the chemical formula to the corresponding chemical reagents.
Comment 17. “2.2. Experimental apparatus” – The names of analytical balance and magnetic stirrer are not necessary.
Answer: Thank you very much for your valuable advice, we have deleted the analytical balance and magnetic stirrer in the Experimental apparatus part.
Comment 18. “To prepare the CCN solution” – Is it a true solution?
Answer: We were sorry that our expression is unscientific. We have revised the expression at the corresponding place.
Comment 19. “The CCN were then redispersed in an appropriate amount of secondary water.” – Which is the “appropriate amount”? What is “secondary water”?
Answer: I am so sorry, the description of secondary water was wrong, the correct answer is deionized water, and we have all revised in the resubmitted manuscript. The obtained CaCO3 NPs were redispersed using 5ml of deionized water.
Comment 20. “Ag-CaCO3 nanoparticles” must be “Ag-CaCO3 nanocomposites” (if Ag particles are in the nano range).
Answer: According to the well-intentioned suggestion of reviewer, the particle size of Ag particles is in the nanometer range, and Ag-CaCO3 nanoparticles have been modified to Ag-CaCO3 nanocomposites.
Comment 21. “an appropriate amount of CCN suspension”, “a suitable volume”, “additional silver nitrate”, etc. – the quantities/ volumes must be provided.
Answer: According to the kind suggestion of the reviewer, we have revised the corresponding parts in the article.
Comment 22. “the mixture was reduced using sodium borohydride” – The whole mixture was reduced?
Answer: Thank you very much for your valuable advice. Here, the mixture referred to Silver nitrate, and we have corrected it in this article. Such as: silver nitrate was immediately reduced using sodium borohydride.
Comment 23. “1 mL of the SCN” – SCN is not a liquid.
Answer: Thank you very much for your valuable advice. We have corrected it in the article.
Comment 24. “a 2 mL solution of calcium lignosulphonate” – the concentration must be given.
Answer: Thank you very much for your good suggestions, we have added the corresponding substance concentration in the article.
Comment 25. “a 2 mL solution of calcium lignosulphonate was determined to be the optimal concentration for the synthesized CCN” – Why “concentration” for a volume?
Answer: According to the well-intentioned suggestion of the reviewer, we have added the corresponding substance concentration in the article because of the improper description before.
Comment 26. Figure 1. – They are different volumes, so different quantities, not “different concentrations”.
Answer: Thank you very much for your good suggestions. We have added the corresponding substance concentration in the article because of the improper description before.
Comment 27. “2 mL of calcium lignosulphonate solution” – the concentration is missing.
Answer: Thank you very much for your valuable advice. Due to the improper description before, we have added the corresponding substance concentration in the article.
Comment 28. “the carbonates were uniformly combined” – What “carbonates”?
Answer: Thank you very much for your valuable advice. We have changed the carbonates to calcium carbonate.
Comment 29. “the optimal concentration of AgNO3 for the synthesis of the SCN was determined to be 3 mL” – “3 mL” is not a “concentration”.
Answer: Thank you very much for your valuable advice. We have revised it to concentration in the article.
Comment 30. Fig. 10 – They are only 4 points, more are needed.
Answer: Thank you very much for your good suggestions. We directly tested forchlorfenuron from 0.0001 mg/mL to 2 mg/mL (6 points) using Ag-CaCO3 nanocomposites as a SERS substrate, the linear equation the SERS intensity of at 988 cm-1 of forchlorfenuron was remarkably correlated linearly with its from 0.01 mg/mL to 2 mg/mL when The limit of detection was determined to be 0.001 mg/mL.
Comment 31. Some examples of inappropriate scientific language: “Characterization analysis”, “Sodium hydrogen carbonate was used to introduce carbonates into the system.”, etc.
Answer: According to the good-will suggestions of the reviewers, we have revised the corresponding parts in the article.
Reviewer #3:
Comment 1. The method used to compare "Effects of different concentrations of calcium lignosulfonate" in Figure 1 is unscientific! It would be more correct for the author to use absorption spectra to perform such experiments.
Answer: We were sorry that our expression is unscientific. We have revised the expression of figure 1, on the other hand, CaCO3 nanoparticles has not UV absorption spectrum.
Comment 2. The scale bars in Figure 2 and Figure 7 are too small, which is very difficult to read and compare.
Answer: Thank you very much for your good suggestions. We have replaced clear scale bars in the revised manuscript.
Figure 2. TEM of the synthesized CaCO3 NPs at different concentrations of calcium lignosulfonate solution(A-E: 0.0004mol/L,0.001mol/L, 0.004mol/L, 0.008mol/L, 0.012mol/L calcium lignosulfonate).
Figure 7. TEM images of the obtained Ag-CaCO3 nanohybrid by adding different amounts of silver nitrate. (A-D: 0 mol/L, 0.002 mol/L, 0.003 mol/L, 0.004 mol/L AgNO3).
Comment 3. The x and y axis coordinates of Figure 10 have disappeared. This will make the reader wonder what data you are presenting.
Answer: Thank you very much for your good suggestions. We have added x and y axis coordinates of Figure 10 in the revised manuscript.
Figure 10. A plot between the SERS intensity at 988 cm-1 and the concentration of forchlorfenuron.
Comment 4. What is the unit of the y-axis coordinate in Figure 6? The "nm" in the x-axis coordinate is also marked very casually!!
Answer: Thank you very much for your valuable advice. We have added appropriate units to the Y axis in Figure 6 and changed the label of the X axis.
Figure 6. UV-visible absorption spectra of the synthesized Ag-CaCO3 nanohybrid under different concentration of AgNO3. (a-d:0.001 mol/L, 0.002 mol/L, 0.003 mol/L, 0.004 mol/L AgNO3; ; e: 0.002 mol/L AgNO3, Calcium carbonate free).
Comment 5. Figure 4 must be added to the standard peaks as a comparison to verify your claim. The current presentation neither analyzes which crystallographic plane nor benchmarks it for comparison.
Answer: Thanks a lot for your valuable advice. We have added the standard card PDF#05-0586 of calcium carbonate to Figure 4 and the corresponding crystal plane.
Figure 4. X-ray diffraction pattern of the synthesized CaCO3 NPs.

Reviewer 3 Report
In the article “Synthesis of Ag-CaCO3 nanoparticles and its application for 2 the detection of forchlorfenuron with SERS”, the authors report the synthesis of Ag-CaCO3 nanoparticles (SCN) using silver nitrate as the precursor solution based on calcium carbonate nanoparticles (CCN). Various techniques including ultraviolet-visible spectrophotometry, transmission electron microscopy, and Raman spectroscopy were used for the characterization analysis of SCN. Their results presented the relatively high SERS activity of SCN, demonstrating it is suitable for the analysis of pesticide residues and the detection of toxic and harmful molecules, thereby contributing to environmental protection. The strategy is interesting but several important information is missing. Therefore, major revision is needed before acceptance. The detailed comments are given below:
1. The method used to compare "Effects of different concentrations of calcium lignosulfonate" in Figure 1 is unscientific! It would be more correct for the author to use absorption spectra to perform such experiments.
2. The scale bars in Figure 2 and Figure 7 are too small, which is very difficult to read and compare.
3. The x and y axis coordinates of Figure 10 have disappeared. This will make the reader wonder what data you are presenting.
4. What is the unit of the y-axis coordinate in Figure 6? The "nm" in the x-axis coordinate is also marked very casually!!
5. Figure 4 must be added to the standard peaks as a comparison to verify your claim. The current presentation neither analyzes which crystallographic plane nor benchmarks it for comparison.
Author Response
Dear reviewers,
On behalf of my co-authors, we thank you very much for giving us an opportunity to revise our manuscript, we appreciate editor and reviewers very much for their positive and constructive comments and suggestions on our manuscript entitled “Fabrication of Ag-CaCO3 nanohybrid for SERS detection of forchlorfenuron” (Molecules-2493104). We have tried our best to revise our manuscript according to the comments, and the revised portion has been marked in red color in the revised manuscript.
The main corrections in the paper and the responds to the reviewer’s comments are shown in the additive list below.
Thank you and looking forward to hearing from you.
Sincerely yours,
Zhihui Luo
Jul 13, 2023
Reviewer #3:
Comment 1. The method used to compare "Effects of different concentrations of calcium lignosulfonate" in Figure 1 is unscientific! It would be more correct for the author to use absorption spectra to perform such experiments.
Answer: We were sorry that our expression is unscientific. We have revised the expression of figure 1, on the other hand, CaCO3 nanoparticles has not UV absorption spectrum.
Comment 2. The scale bars in Figure 2 and Figure 7 are too small, which is very difficult to read and compare.
Answer: Thank you very much for your good suggestions. We have replaced clear scale bars in the revised manuscript.
Figure 2. TEM of the synthesized CaCO3 NPs at different concentrations of calcium lignosulfonate solution(A-E: 0.0004mol/L,0.001mol/L, 0.004mol/L, 0.008mol/L, 0.012mol/L calcium lignosulfonate).
Figure 7. TEM images of the obtained Ag-CaCO3 nanohybrid by adding different amounts of silver nitrate. (A-D: 0 mol/L, 0.002 mol/L, 0.003 mol/L, 0.004 mol/L AgNO3).
Comment 3. The x and y axis coordinates of Figure 10 have disappeared. This will make the reader wonder what data you are presenting.
Answer: Thank you very much for your good suggestions. We have added x and y axis coordinates of Figure 10 in the revised manuscript.
Figure 10. A plot between the SERS intensity at 988 cm-1 and the concentration of forchlorfenuron.
Comment 4. What is the unit of the y-axis coordinate in Figure 6? The "nm" in the x-axis coordinate is also marked very casually!!
Answer: Thank you very much for your valuable advice. We have added appropriate units to the Y axis in Figure 6 and changed the label of the X axis.
Figure 6. UV-visible absorption spectra of the synthesized Ag-CaCO3 nanohybrid under different concentration of AgNO3. (a-d:0.001 mol/L, 0.002 mol/L, 0.003 mol/L, 0.004 mol/L AgNO3; ; e: 0.002 mol/L AgNO3, Calcium carbonate free).
Comment 5. Figure 4 must be added to the standard peaks as a comparison to verify your claim. The current presentation neither analyzes which crystallographic plane nor benchmarks it for comparison.
Answer: Thanks a lot for your valuable advice. We have added the standard card PDF#05-0586 of calcium carbonate to Figure 4 and the corresponding crystal plane.
Figure 4. X-ray diffraction pattern of the synthesized CaCO3 NPs.

Round 2
Reviewer 3 Report
The revised version has been modified according to my suggestions. I recommend it for acceptance for publication.
Author Response
Dear reviewer,
Thank you for your good suggestions, it makes our work better.
Kind regards
Zhihui Luo
Jul 23,2023